# Bone marrow imaging reveals the migration dynamics of neonatal hematopoietic stem cells

Yuji Takihara[1,2], Takumi Higaki [3,4], Tomomasa Yokomizo[5], Terumasa Umemoto [5], Kazunori Ariyoshi[5], Michihiro Hashimoto[5], Maiko Sezaki[5], Hitoshi Takizawa [1,5], Toshihiro Inoue[1], Toshio Suda [2,5✉] & Hidenobu Mizuno [1,5✉]

Hematopoietic stem cells (HSCs) are produced from the blood vessel walls and circulate in the blood during the perinatal period. However, the migration dynamics of how HSCs enter the bone marrow remain elusive. To observe the dynamics of HSCs over time, the present study develops an intravital imaging method to visualize bone marrow in neonatal long bones formed by endochondral ossification which is essential for HSC *niche* formation. Endogenous HSCs are labeled with tdTomato under the control of an HSC marker gene *Hlf*, and a customized imaging system with a bone penetrating laser is developed for intravital imaging of tdTomato-labeled neonatal HSCs in undrilled tibia, which is essential to avoid bleeding from fragile neonatal tibia by bone drilling. The migration speed of neonatal HSCs is higher than that of adult HSCs. Neonatal HSCs migrate from outside to inside the tibia via the blood vessels that penetrate the bone, which is a transient structure during the neonatal period, and settle on the blood vessel wall in the bone marrow. The results obtained from direct observations in vivo reveal the motile dynamics and colonization process of neonatal HSCs during bone marrow formation.

[1] Graduate School of Medical Sciences, Kumamoto University, 1-1-1 Honjo, Chuo-ku, Kumamoto, Japan. [2] Cancer Science Institute of Singapore, National University of Singapore, 14 Medical Drive, #12-01, 117599 Singapore, Singapore. [3] Graduate School of Science and Technology, Kumamoto University, 2-39-1 Kurokami, Chuo-ku, Kumamoto, Japan. [4] International Research Organization for Advanced Science and Technology (IROAST), Kumamoto University, 2-39-1 Kurokami, Chuo-ku, Kumamoto, Japan. [5] International Research Center for Medical Sciences (IRCMS), Kumamoto University, 2-2-1 Honjo, Chuo-ku, Kumamoto, Japan. ✉email: csits@nus.edu.sg; hmizuno@kumamoto-u.ac.jp

Hematopoietic stem cells (HSCs) are characterized by multilineage differentiation and the capacity for self-renewal to sustain lifelong hematopoiesis[1,2]. HSCs actively divide with a high metabolic status in the neonatal stage[3,4] and then show a quiescent state with decreased metabolic status to maintain HSCs by preventing excess cell division in the adult stage[5]. Transcriptional HSC profiles are dramatically changed during the perinatal period[6] and these alternations to HSC status could be related to their microenvironment, known as their *niche*[7], during development. Previous studies have suggested that HSC precursors are generated from the aorta–gonad–mesonephros region during the embryonic period and then transfer to the fetal liver, where they mature into definitive HSCs. HSCs are thought to migrate to the bone marrow *niche* during the perinatal period[8,9]. However, the migration dynamics of HSCs during early development remain unclear in living animals.

The dynamics of HSCs have been studied in adult mice using intravital imaging. Fluorescent-labeled HSCs have been transplanted into irradiated mice to observe HSC dynamics in vivo, and intravital imaging of the bone marrow has been performed in the calvarium[10–13] or drilled long bones[14–17]. However, irradiation prior to transplantation could affect the dynamics of HSCs by changing the surrounding microenvironment[18–20]. Therefore, transplantation of large numbers of HSCs into mice that have not been irradiated was conducted to avoid the irradiation effect[21]. Furthermore, reporter mice have been developed to selectively label HSCs. Long bone imaging after optical clearing in α-catulin[GFP] mice suggests that HSCs mainly reside in the perisinusoidal *niche*[22]. The Mds1[GFP/+]Flt3[Cre] system allows accurate identification of long-term HSCs through deletion of green fluorescent protein (GFP) in progenitor cells, which enables intravital imaging of endogenous long-term HSCs in the calvarium bone marrow in adult mice[10].

Despite over a decade of years of research into intravital imaging of bone marrow, the dynamics of HSCs in the neonatal stage remain unexplored. The present study established a method for imaging endogenous HSCs within the bone marrow of tibial bones in adult and neonatal mice without damaging the bone. The tibial bone is formed by endochondral ossification, which is important for the formation of the HSC *niche*[23], although the calvarium is formed via intramembranous ossification[24]. Therefore, we focused on imaging the bone marrow of the tibia rather than that of the calvarium. Hlf-tdTomato knock-in (KI) mice[25], in which a red fluorescent protein tdTomato is expressed under the promoter control of the HSC marker gene *Hlf*[26,27], was used for intravital imaging of endogenous HSCs in the tibia. The findings of the present study revealed the changes in the dynamics of HSCs within the bone marrow between the adult and neonatal stages and the process of HSC homing to the bone marrow during the neonatal period.

## Results

### Cells with the highest Hlf-tdTomato expression levels have bone marrow reconstitution capacity.

We previously reported that HSCs have higher levels of tdTomato than other hematopoietic progenitors in the fetal livers of Hlf-tdTomato KI mice[25]. Transplantation experiments were performed to confirm the stem cell potential of bone marrow cells with higher levels of Hlf-tdTomato in the bone marrow of the tibia. Flow cytometry analysis showed that $47.0 \pm 4.4\%$ of whole bone marrow cells from adult long bones of Hlf-tdTomato KI mice were positive for CD45, which is a panhematopoietic marker (Fig. 1a, left panel, $n = 4$). Cells with the top 0.011% tdTomato intensity within the CD45-positive cells (0.0049% of the whole bone marrow cells) were defined as Hlf-tdTomato[hi] cells (Fig. 1b). All the Hlf-

tdTomato[hi] cells were located within the Sca-1$^+$c-Kit$^+$ fraction, including phenotypic HSCs (CD150$^+$CD48$^-$) and short-term HSCs (CD150$^-$CD48$^-$) (Fig. 1a, right panel and Supplementary Fig. 1). To determine whether Hlf-tdTomato[hi] cells show a high frequency of functional HSCs within the bone marrow of the tibia, we compared engraftment capacity between the Hlf-tdTomato[hi] and Hlf-tdTomato$^-$ cells (44.3% of the whole bone marrow cells) by the transplantation assay (Fig. 1b). A total of 100 Hlf-tdTomato[hi] cells were capable of engraftment after primary and secondary transplantation, whereas 5000 Hlf-tdTomato$^-$ cells were not capable (Fig. 1c, d). These results indicate that functional HSCs were enriched within the Hlf-tdTomato[hi] fraction in the bone marrow of the long bones.

### Intravital imaging enables observation of endogenous HSCs in the bone marrow of undrilled long bones.

Next, we developed a method to observe the dynamics of Hlf-tdTomato[hi] HSCs in the tibial bone marrow in vivo (Fig. 2). Previous studies have used drilled tibia for intravital imaging of HSCs in the long bones[14–17]. However, drilling of the long bones precluded the comparison of the HSC dynamics between adults and neonates for two reasons. First, drilling could disturb the microenvironment of the bone marrow, and second, the long bone of neonates is too fragile to be drilled, and it is not possible to avoid bleeding from the blood vessels penetrating the neonatal tibia (Supplementary Fig. 2a and Supplementary Movie 1). A multiphoton imaging system equipped with a bone-penetrating fiber laser (average power, >2 W; pulse width, 55> fs; wavelength, 1070 nm) was established to overcome the limitations of the conventional methods (Fig. 2a). In our system, tdTomato-positive cells were observed under the intact tibial bone tissue, which was visualized with second harmonic generation (SHG) signals in adult mice (3 months old) in vivo (Fig. 2b; Supplementary Movie 2). Blood capillaries in the bone marrow were visualized by intravenous injection of an infrared fluorescent dye Qtracker 655 to confirm the location of the tdTomato-positive cells within the tibial bone marrow. Intravital imaging showed that tdTomato-positive cells were located around the blood capillary network (Fig. 2c, d), which are typical blood vessel patterns in the bone marrow of long bones[28]. These results suggest that Hlf-tdTomato-positive cells in the intact tibial bone marrow can be observed by the method developed in the present study.

The technical advances of our method were evaluated by comparing it with the conventional method. For the conventional method, intravital imaging of Runx1-GFP transgenic mice, in which HSCs and progenitor cells strongly express GFP[29], was conducted using a short wavelength (920 nm) laser which does not easily penetrate bone. Intravital imaging showed that Runx1-GFP labeled cells in the bone marrow were blurred without drilling (Fig. 2e), and GFP signals were only clearly observed after drilling (Fig. 2f), indicating that drilling is essential for the conventional method. As previously reported[11], artificial background signals were observed in the green channel (Fig. 2f), whereas these were rarely seen using our imaging method (Fig. 2b). These results highlight the advances of the intravital imaging developed in the present study.

There were $18.2 \pm 1.0$ tdTomato-positive cells in the volume of the intravital images (around 600 μm × 600 μm × 100 μm = $3.6 \times 10^7$ μm$^3$; $n = 5$ volumes from five mice). Immunohistochemical staining of the tibial bone sections obtained from Hlf-tdTomato KI mice was conducted (Fig. 3a) to determine which tdTomato-positive cells in the intravital images corresponded to the Hlf-tdTomato[hi] HSCs that were identified in the flow cytometry analysis (Fig. 1). In the histological sections of the tibial bone, there were $18,201 \pm 933$ CD45-positive cells and

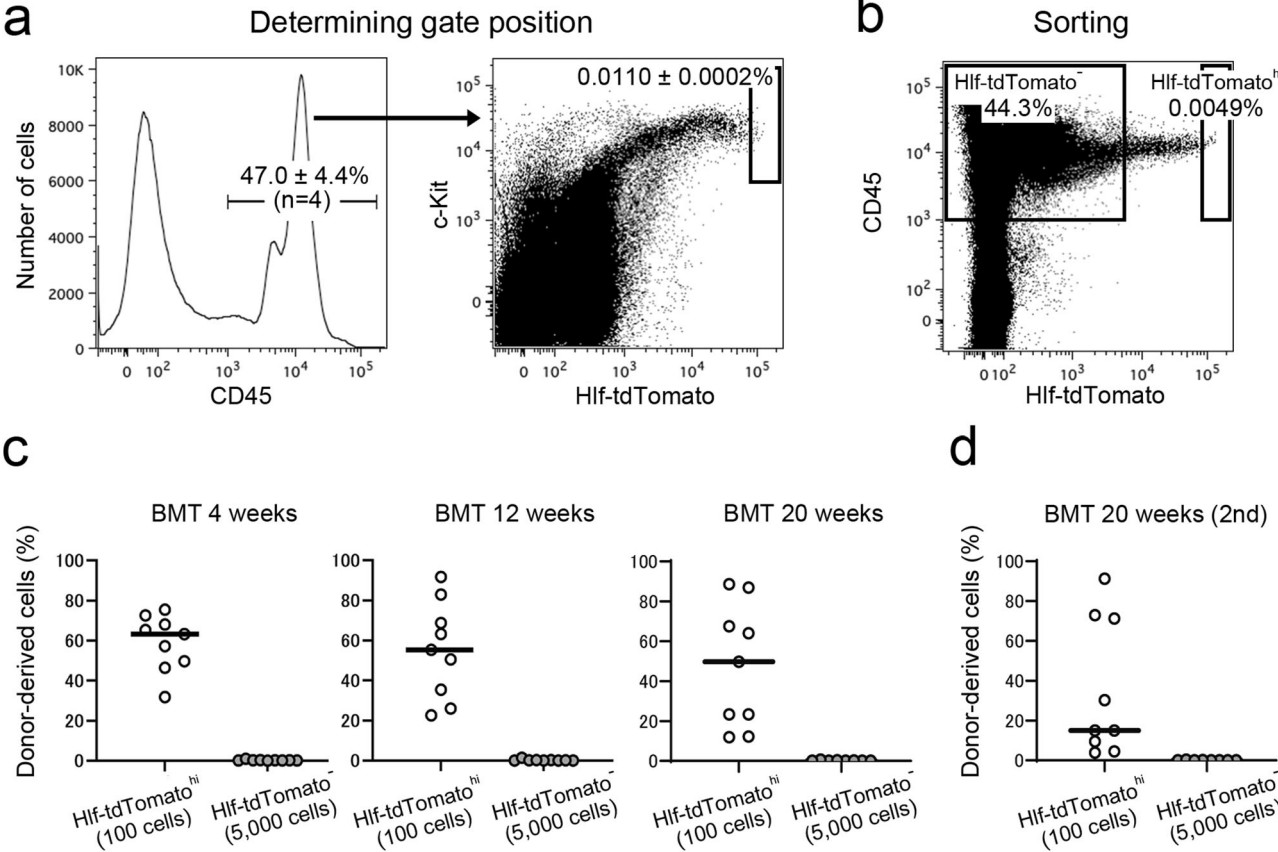

**Fig. 1 Transplantation experiments showing stem cell potential of tdTomato$^{hi}$ cells. a** Hlf-tdTomato expression in the bone marrow blood cells obtained from long bones. The black box indicates the Hlf-tdTomato$^{hi}$ population. **b** Flow cytometry sorting of Hlf-tdTomato cells. The Hlf-tdTomato$^{hi}$ box population was used for transplantation experiments. The tdTomato$^−$ box population was used as a negative control. **c** Bone marrow transplantation (BMT) experiments. Irradiated mice were transplanted with 100 tdTomato$^{hi}$/CD45$^+$ cells or 5000 tdTomato$^−$/CD45$^+$ cells. **d** Second BMT experiments.

1,630 ± 269 tdTomato-positive cells within the same volume as the in vivo-imaged volume ($n = 3$ sections from three mice; Fig. 3b, c). Since the cells with the top 0.011% tdTomato intensity in CD45-positive cells were defined as Hlf-tdTomato$^{hi}$ cells (Fig. 1), those with the top two tdTomato fluorescent intensities in the in vivo images corresponded to Hlf-tdTomato$^{hi}$ cells (18,201 cells × 0.011% = 2 cells; Fig. 3c). We defined the remaining Hlf-tdTomato-positive cells with lower intensity in the intravital images that include differentiated progenitor cells[25] as Hlf-tdTomato$^{low}$ cells (16 cells in Fig. 3c), which had lower stem cell potential (Supplementary Fig. 3). Most of the tdTomato-positive cells in the histological sections were not visible using intravital imaging due to the very low intensity (1612 cells in Fig. 3c). These results suggested that the cells with top two tdTomato fluorescent intensities in the in vivo image were HSCs.

**Intravital imaging of intact tibia reveals endogenous HSCs are stationary in adult bone marrow.** Three-dimensional time-lapse imaging of undrilled tibial bone marrow was performed to observe the in vivo dynamics of Hlf-tdTomato$^{hi}$ HSCs (Fig. 4a and Supplementary Movie 3). Artifactual movement of the image area, mainly caused by the heartbeat, was corrected using image registration. Hlf-tdTomato$^{hi}$ HSCs were stationary (Fig. 4b), although they showed oscillatory movements in the restricted area. In contrast, Hlf-tdTomato$^{low}$ cells migrated (Fig. 4c). Quantitative analysis of the HSC migration using TrackMate[30] revealed that the velocity of the Hlf-tdTomato$^{hi}$ HSCs (0.096 ± 0.019 μm/min, 10 cells from five mice) was significantly

lower than that of Hlf-tdTomato$^{low}$ cells (0.169 ± 0.017 μm/min, 81 cells from five mice; $p = 0.008$; $t = 2.886$; $g = 0.505$; Fig. 4d). We also showed long-term engraftment of Hlf-tdTomato$^{hi}$ cells (Fig. 1c, d and Supplementary Fig. 1), indicating that HSCs were stationary but oscillatory. However, differentiated cells were motile in the bone marrow of adult long bones. Therefore, our findings demonstrate that Hlf-tdTomato KI mice, the "inside-bone intravital imaging system" and quantitative bioimaging analysis facilitate the evaluation of the migration dynamics of endogenous HSCs in the native microenvironment of long bones.

**Changes in expression of cell migration-related genes in neonatal HSCs.** Neonatal HSCs are characterized by fast cell cycling and higher mitochondrial membrane potential[4], indicating changes in the cellular properties between adult and neonatal HSCs. Gene expression patterns were compared between developing HSCs and matured HSCs using RNA-seq to evaluate changes in the properties of neonatal HSCs.

Gene set enrichment analysis (GSEA) showed significant enrichment in cell migration-related genes in neonatal HSCs (Supplementary Fig. 4a, b). Consistently, changes in the expression of genes related to the cytoskeleton and cell adhesion were observed in neonatal HSCs (Supplementary Fig. 4c–f). Differences in the cell migration-related genes indicated differences in the migration dynamics of neonatal and adult HSCs in the tibial bone marrow. From these results, we focused on the migration dynamics in subsequent experiments for intravital imaging of neonatal mice.

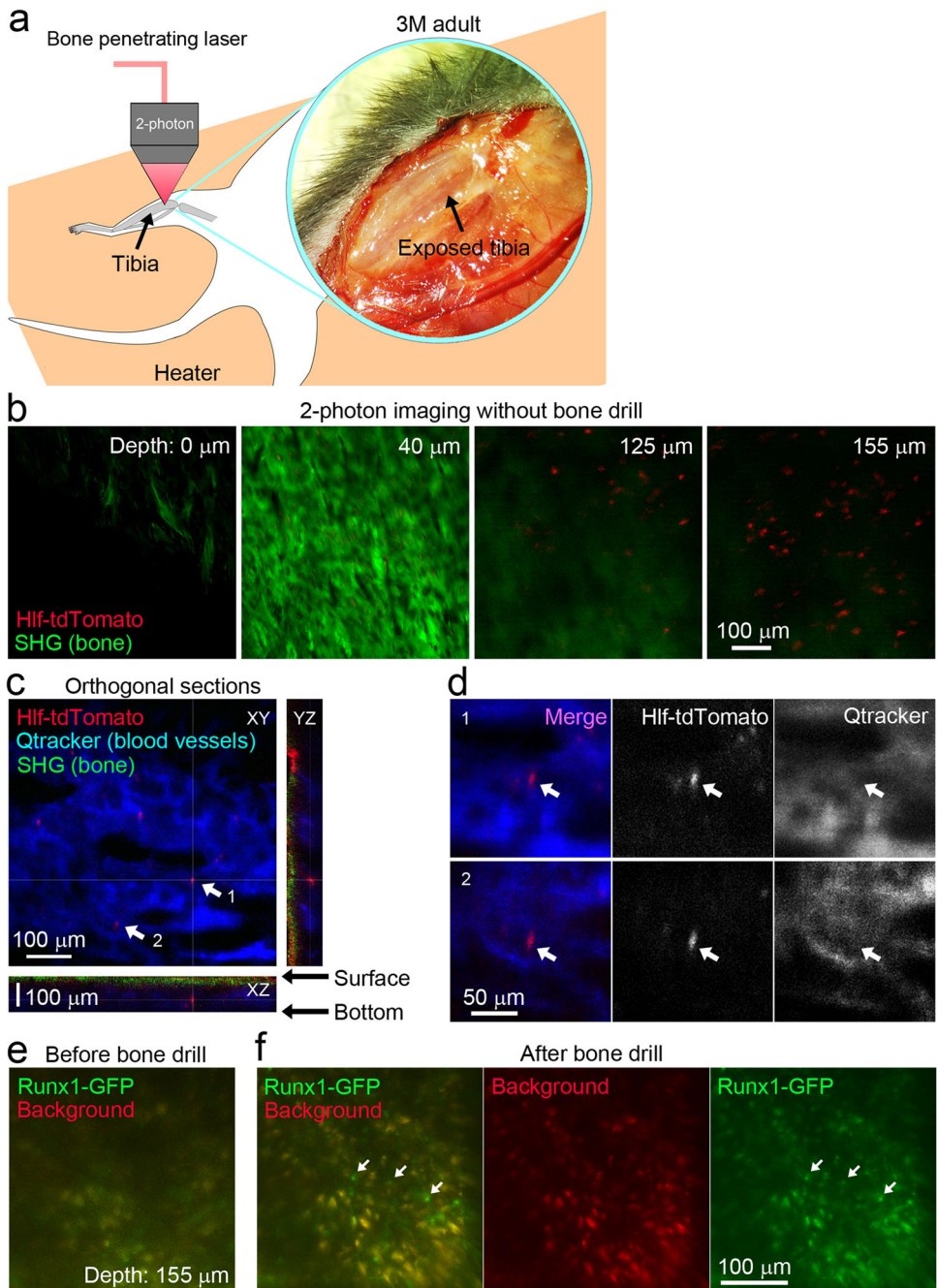

**Fig. 2 Intravital two-photon imaging of Hlf-tdTomato-labeled cells in tibial bone marrow. a** Experimental schema of intravital imaging of the tibial bone. The tibial bone was exposed, and the bone marrow was imaged without bone drilling using a high-powered infrared laser. **b** Z-stack images obtained via in vivo imaging of the bone marrow in the undrilled tibia of Hlf-tdTomato KI mice. Hlf-tdTomato-positive cells were observed in the bone marrow cavity. Bone surface, bone, and bone marrow cavities. The depth is indicated in the upper right corner of each panel. SHG, second harmonic generation. **c** Orthogonal view of 3D images confirmed that Hlf-tdTomato-positive cells were in the bone marrow under the tibial bone. Blood capillaries in the tibial bone marrow were visualized by intravenous injection of Qtracker 655. Cells indicated with arrows 1 and 2 are shown in higher magnification in (**d**). See also Supplementary Movie 2. **d** Higher magnification images of Hlf-tdTomato-positive cells located around the bone marrow capillary. **e** Intravital imaging of a Runx1-GFP mouse before drilling. Depth is 155 μm from the bone surface. **f** Intravital imaging of the Runx1-GFP mouse after drilling. The same region in (**d**) was reimaged after drilling. Background signals were imaged in the red channel, suggesting that most signals in the green channel were artificial backgrounds. Arrows indicate Runx1-GFP cells that had no background signals in the red channel.

**Migration of Hlf-tdTomato^hi cells in the neonatal tibial bone marrow and bone cavity.** Migration of HSCs into the bone marrow from other hematopoietic organs has been hypothesized since adult-type definitive HSCs are generated from the aorta–gonad–mesonephros region[8,25]. However, the migration dynamics of HSCs during development is unclear. Therefore,

three-dimensional time-lapse imaging of the bone marrow was performed in the undrilled tibia in neonates (postnatal day 2; Fig. 5a, left). The orthogonal view confirmed that the intravital imaging of the intact tibial bone marrow enabled the observation of endogenous tdTomato-positive cells in neonatal Hlf-tdTomato KI mice (Fig. 5a, right; Supplementary Movie 4). Quantitative

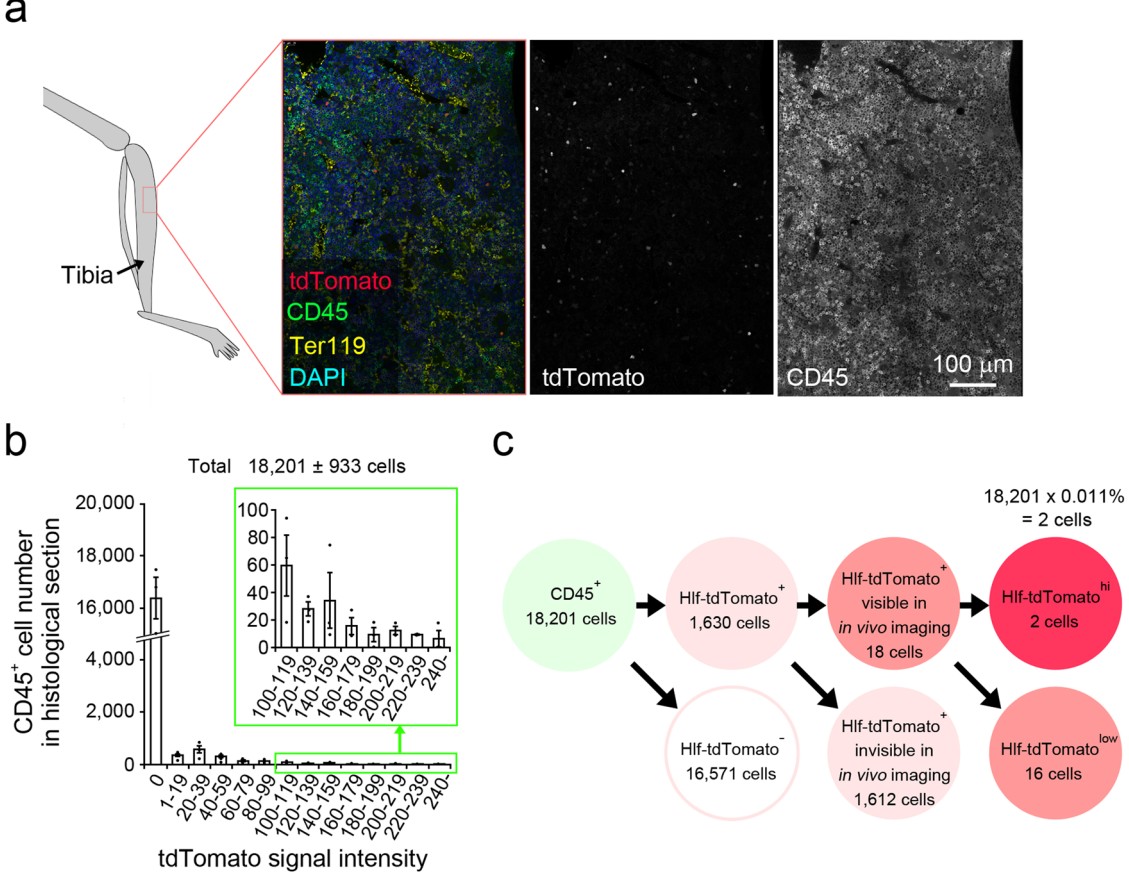

**Fig. 3 Identification of Hlf-tdTomato$^{hi}$ cells in the intravital images. a** Immunohistochemical staining of tibial bone sections was obtained from an adult Hlf-tdTomato KI mouse. Ter-119, red blood cell marker; CD45, blood cell marker except for mature red blood cells and platelets; DAPI, nuclear marker. **b** Fluorescence distribution of tdTomato signals in CD45-positive cells in the histological sections of the tibial bone. The number of CD45-positive cells within the same volume as the in vivo-imaged volume was 18,201 ± 933 cells. Error bars indicate standard error. **c** The number of CD45 and tdTomato-double-positive cells within the same volume as the in vivo-imaged volume was 1630 ± 269 cells. The number of tdTomato-positive cells was 18.2 ± 1.0 cells, suggesting that the remaining 1612 cells were not visible in the intravital images due to the very low fluorescent intensity. The cells with the top 0.011% tdTomato intensity in CD45-positive cells were defined as Hlf-tdTomato$^{hi}$ cells (Fig. 1); therefore, the cells with the top two tdTomato fluorescent intensities in the in vivo images correspond to the Hlf-tdTomato$^{hi}$ cells (18,201 cells × 0.011% = 2 cells). We defined the remaining Hlf-tdTomato-positive cells with lower fluorescent intensity (16 cells in average) as Hlf-tdTomato$^{low}$ cells for quantitative analysis of Hlf-tdTomato-positive cell dynamics in Figs. 4 and 5.

analyses showed that the velocity of Hlf-tdTomato$^{hi}$ cells (1.516 ± 1.010 μm/min, six cells from three mice; top two fluorescent intensities) was much higher than that of Hlf-tdTomato$^{low}$ cells (0.078 ± 0.010 μm/min, 24 cells from three mice; $p = 0.002$; $Z = 3.059$; $r = 0.558$) in neonates (Fig. 5b). Furthermore, the velocity of Hlf-tdTomato$^{hi}$ cells in neonates was much higher than that in adults ($p = 0.017$; $Z = 2.386$; $r = 0.597$, Supplementary Fig. 2b). These results suggest that HSCs are motile in the tibial bone marrow of neonates.

Intravital imaging was conducted after visualizing blood vessels by injecting Qtracker 655 via the superficial temporal vein[31] to check whether the motile Hlf-tdTomato$^{hi}$ cells were extravascular or intravascular. Some of the Hlf-tdTomato$^{hi}$ cells rapidly migrated in the blood vessels (Fig. 5c, d, white arrows). Moreover, Hlf-tdTomato$^{hi}$ cell attached to the vessels during the imaging session, indicating extravasation and homing to the bone marrow (Fig. 5c, 63 and 100 min; Supplementary Movie 5). In contrast, Hlf-tdTomato$^{hi}$ cells located outside the capillaries, which appeared to be extravasated prior to the imaging session, were stationary (Fig. 5c, yellow arrows). These results suggest that motile Hlf-tdTomato$^{hi}$ cells migrate in the blood vessels of the neonatal tibia.

Finally, migration of HSCs from outside the bone marrow was observed using time-lapse imaging of the tibial bone cavity, where the blood vessels that penetrate the bone are located (Fig. 6a). An Hlf-tdTomato$^{hi}$ cell was observed to rapidly migrate within the bone cavities (Fig. 6b and Supplementary Movie 6). Interestingly, the distance between the observed cell and the inner bone surface appeared to be increased (Supplementary Fig. 5a, b), suggesting migration of the cell to the deeper part of the bone. Taken together, these results indicate that HSCs migrate in the bone cavities and bone marrow during tibial bone marrow formation.

**Discussion**

An intravital imaging method for endogenous HSCs in the bone marrow of long bones was established without drilling using a customized multiphoton imaging system. This system was used to reveal that HSCs within the bone marrow of long bones in adult mice are stationary but oscillatory. In contrast, HSCs in neonatal mice were found to be motile within the bone marrow of the long bones and showed differential dynamic behavior from adult HSCs. GSEA consistently demonstrated enrichment of cell migration-related genes in neonatal HSCs. Neonatal HSCs

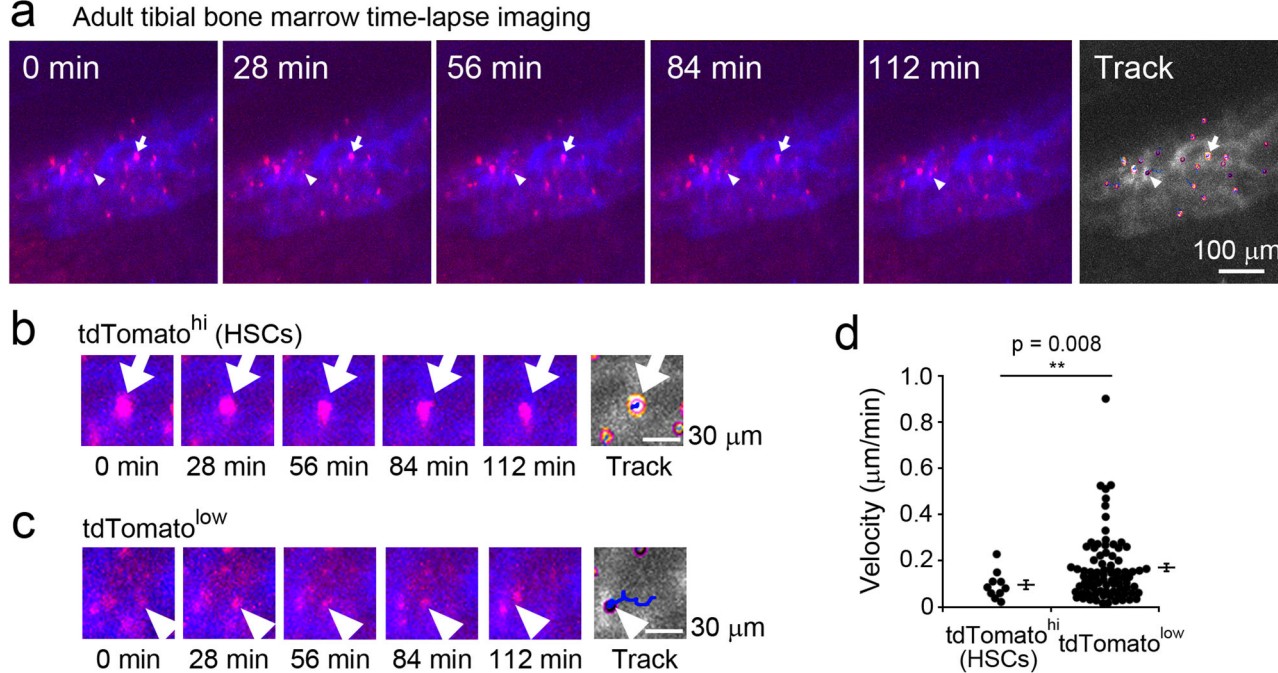

**Fig. 4 Time-lapse imaging of Hlf-tdTomato-labeled cells in the tibial bone marrow. a** Hlf-tdTomato-positive cells in the tibial bone marrow were imaged for 2 h in vivo. Arrows indicate Hlf-tdTomato[hi] HSCs and arrowheads indicate Hlf-tdTomato[low] cells. See also Supplementary Movie 3. **b** Higher magnification time-lapse images of Hlf-tdTomato[hi] HSCs. **c** Higher magnification time-lapse images of Hlf-tdTomato[low] cells. **d** Quantitative comparison of the migration dynamics between Hlf-tdTomato[hi] cells (ten cells from five mice) and Hlf-tdTomato[low] cells (81 cells from five mice).

migrate to the bone marrow via vessels that penetrate the cavities in immature tibial bone (Fig. 6c), which rarely exist in the adult tibia (Fig. 2b). Thus, the results of the present study provide evidence for HSC migration to the bone marrow at the early stage of lifelong hematopoiesis before HSCs become quiescent in the cell cycle, which has been hypothesized[8] but has not been directly verified in living animals.

The intravital imaging described in the present study enables the observation of endogenous HSCs in native microenvironments of undrilled long bones (Figs. 2, 4, 5, 6 and Supplementary Movies 2–6). This imaging approach has two important advantages. First, intravital imaging of bone marrow in the long bones does not require drilling. To date, previous studies have reported intravital imaging of bone marrow in the calvarium and drilled long bones[10–13,16]. Although intravital imaging of bone marrow in the calvarium can be achieved without drilling, the calvarium is formed by intramembranous ossification. Endochondral ossification is required for the formation of the HSC *niche* and progenitors from the calvarium do not efficiently form an HSC *niche*;[23] therefore, intravital imaging in long bones is essential to evaluate neonatal HSC migration dynamics in native microenvironments. Furthermore, it was necessary to establish intravital imaging of long bones without drilling in neonates because bleeding is unavoidable when neonatal long bones are drilled (Supplementary Fig. 2a and Supplementary Movie 1). In the present study, a bone-penetrating laser (high power at a long-wavelength) for a red fluorescent protein tdTomato enabled imaging of HSCs in the bone marrow without bone drilling in vivo (Fig. 2 and Supplementary Movie 2). The advantage of our approach is shown by the clearer images and much lower background signals of Hlf-tdTomato[hi] cells (Fig. 2b–d), compared with those of GFP-positive cells observed by intravital imaging of Runx1-GFP transgenic mice (Fig. 2e, f). The second advantage is the observation of endogenous HSCs, which is enabled by Hlf-tdTomato KI mice. Previous studies have mainly analyzed the dynamics of transplanted HSCs. However, it is unclear whether

the dynamics of exogenous HSCs injected into irradiated mice is the same as those of endogenous HSCs in a steady-state[18–20]. In the present study, HSC dynamics were observed in native microenvironments of neonatal bone marrow without radiation. Compared with imaging of endogenous HSCs using α-catulin[GFP] mice after bone clearing[22] and intravital imaging of endogenous HSCs in the bone marrow of calvarium by Mds1[GFP/+]Flt3[Cre] system[10], our imaging approach is unique in intravital imaging of endogenous HSCs in undrilled long bones, which is essential for studying the dynamics of HSCs in neonates.

Intravital imaging has revealed a biological characteristic, namely motile dynamics, of neonatal HSCs (Fig. 5, Supplementary Fig. 2b, and Supplementary Movies 4–6) in addition to neonatal HSC properties, such as active cell cycling and high metabolic status. The migration speed of neonatal Hlf-tdTomato[hi] cells was higher than that of adult Hlf-tdTomato[hi] cells, and neonatal Hlf-tdTomato[hi] cells included cells migrating in blood vessels. Interestingly, the velocity of an Hlf-tdTomato[hi] cell (Fig. 5b) was much slower than the velocity of blood flow in the calvarial sinusoids[32]. Therefore, neonatal HSCs may not passively flow in the vessels but may actively home to the bone marrow vessels. Taken together with the RNA-seq results in the present study regarding differences in expression of gene related to cell migration, cytoskeleton, and cell adhesion between developing and mature HSCs (Supplementary Fig. 4), these results suggest that neonatal HSCs have different dynamic properties from adult HSCs to migrate between distinct hematopoietic organs during development. On the other hand, adult HSCs in the bone marrow of long bones were stationary but shown to oscillate in the present study (Fig. 4 and Supplementary Movie 3). In agreement with this finding, HSCs from adult mice without infection have lower linear progression (displacement/total track length) than those with acute infection, whereas adult HSCs oscillate in the bone marrow of irradiated recipient mice[13]. HSC oscillation may reflect dynamic interaction between HSCs and *niche* cells.

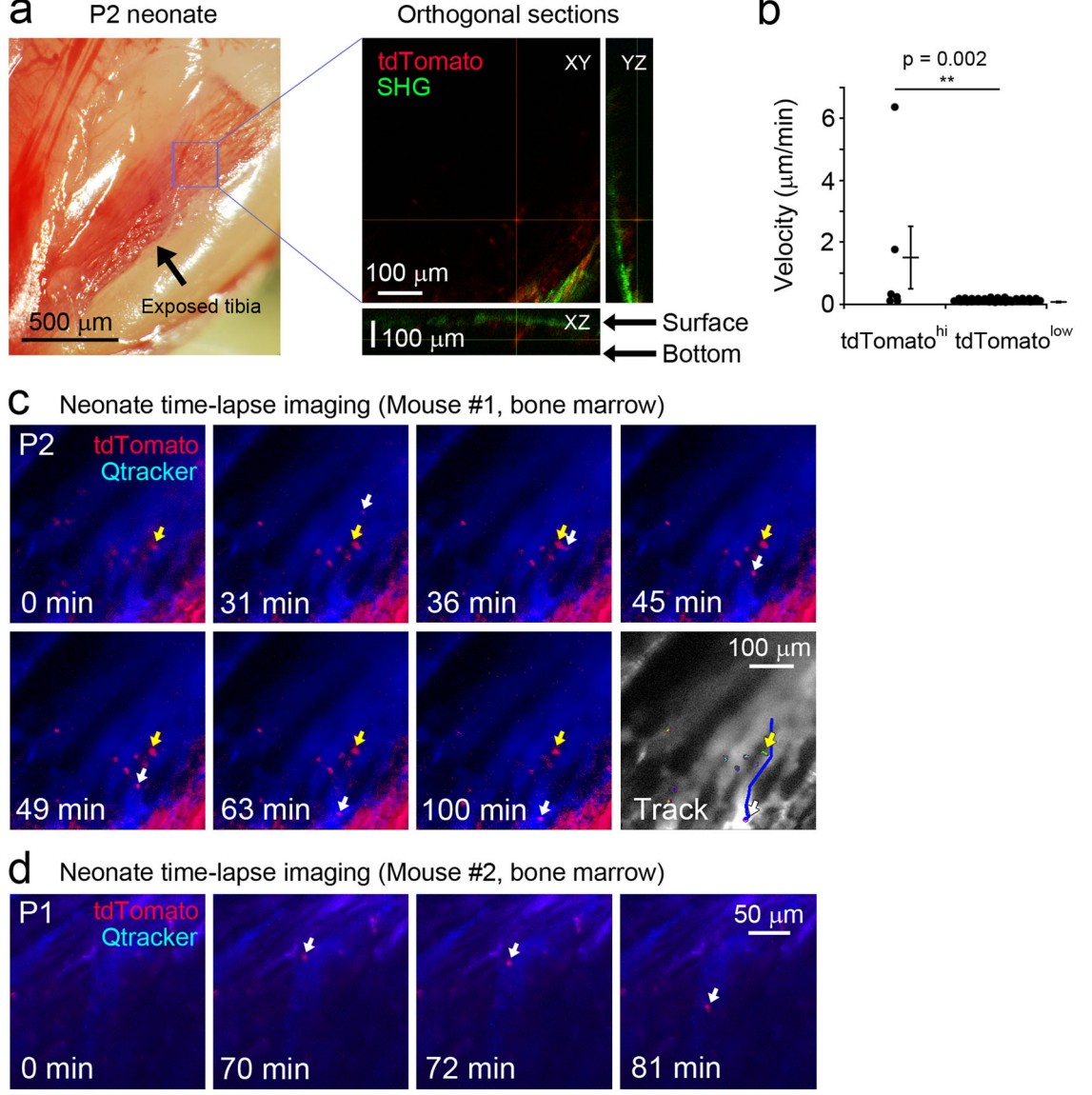

**Fig. 5 Intravital time-lapse imaging of Hlf-tdTomato-labeled cells in the neonatal tibial bone marrow. a** Tibial bone was exposed on postnatal day 2 (left). Orthogonal view image of the tibial bone marrow (right). See also Supplementary Movie 4. **b** Quantitative comparison of the migration dynamics between Hlf-tdTomato[hi] cells (six cells from three mice) and Hlf-tdTomato[low] cells (24 cells from three mice) in neonates. **c** Time-lapse images of developing (P2) bone marrow showing migration of an Hlf-tdTomato[hi] cell in the blood vessels. White arrows indicate migrating cells in the blood vessel and yellow arrows indicate stable cells outside the blood vessels. See also Supplementary Movie 5. **d** Another example of an Hlf-tdTomato[hi] cell migrating in the bone marrow blood vessels in a neonatal mouse (P1).

Intravital imaging showed that the tibias of neonates contained holes through which blood vessels passed (Fig. 6a), and neonatal HSCs migrated through the blood vessels and settled on the blood vessels in the bone marrow (Figs. 5c, d, 6b, Supplementary Fig. 5, and Supplementary Movies 5, 6). In contrast, intravital imaging demonstrated that such holes and blood vessels penetrating the bones rarely exist in the matured tibia of adults (Fig. 2b). From these results, we propose a "transient vessel penetration model" for HSC homing during bone marrow formation (Fig. 6c). In mice, long bones have perfused vasculatures at around embryonic day 16.5 after endochondral ossification[31]. Transient blood vessels penetrating the immature long bones and bone marrow *niche* formed after endochondral ossification may confer an optimal environment for HSC migration into the bone marrow from other hematopoietic organs. Similarly, in mouse liver, the

transition of portal vessels from Neuropilin-1[+]Ephrin-B2[+] artery to EphB4[+] vein phenotype at birth is associated with HSC release from the portal vessels[7].

In conclusion, we developed an intravital imaging method to examine endogenous HSCs in the native microenvironments of the bone marrow without drilling long bones formed by endochondral ossification, which is essential for HSC *niche* formation. The next challenge is to increase the success rate of neonatal imaging. This can be overcome by improving the intravenous injection method required for blood vessel visualization, and refining the imaging conditions such as anesthesia. The strategy would further elucidate the dynamics of endogenous HSC migration between native *niches* in distinct hematopoietic organs during development and the mechanisms of HSC migration abnormalities in diseases.

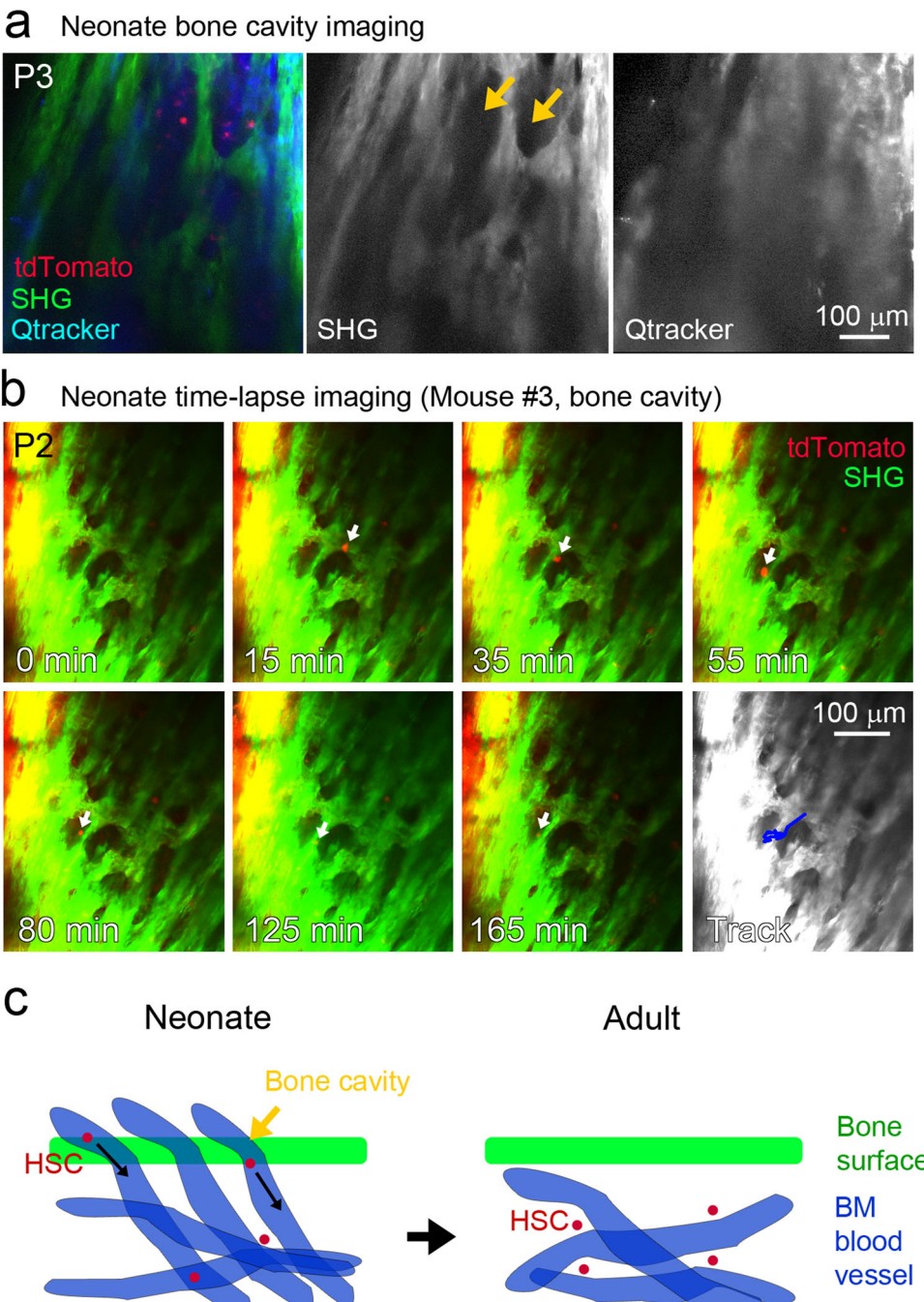

**Fig. 6 Intravital time-lapse imaging of Hlf-tdTomato-labeled cells in the bone cavity of the neonatal tibia. a** Two-photon images showing the blood vessels penetrating the bone cavity in the tibial bone in a neonatal mouse (P3). Yellow arrows indicate bone cavities. **b** Time-lapse images of Hlf-tdTomato[hi] cells migrating from the bone surface to the bone marrow cavity. White arrows indicate a migrating cell in the bone cavity. See also Supplementary Movie 6. **c** "Transient vessel penetration model" for HSC homing during bone marrow formation. HSCs migrate from outside the tibia to inside via transient blood vessels penetrating the bone during bone marrow formation.

## Methods

**Animals**. Hlf-tdTomato KI mice were generated in the previously study[25]. Runx1-GFP mice were kindly provided by M. Osato (Cancer Science Institute of Singapore, National University of Singapore)[29]. C57BL/6-Ly5.1 mice were used for competitive bone marrow transplantation experiments. Data were taken from distinct mice or cells unless otherwise noted. All animal experiments were conducted according to the guidelines of Kumamoto University on animal use.

**Flow cytometry**. For flow cytometric analysis, bone marrow cells were prepared by crushing the bones using a mortar and pestle[25]. Antibodies against CD45 (30-F11,

1:100, BioLegend), c-Kit (2B8, 1:400, BioLegend), CD45.1 (A20, 1:200, BioLegend), and CD45.2 (104, 1:200, Biolegend) were used for flow cytometric analysis. Flow cytometric analysis and cell sorting were conducted on BD FACS Aria III cell sorter (BD Biosciences).

**Transplantation**. Either 100 tdTomato[hi] or 5000 tdTomato[−] cells obtained from Hlf-tdTomato KI mice (Ly5.1/Ly5.2) were intravenously transplanted into lethally irradiated C57BL/6 (Ly5.2) mice (total 10 Gy) along with Ly5.1 whole bone marrow cells ($2 \times 10^5$ cells) as competitor cells. Chimerism in the peripheral blood of the recipients was analyzed by flow cytometry at the indicated time points[33]. For Fig. 1c, the peripheral blood was obtained from the same set of mice.

**Intravital tibial bone marrow imaging**. Hlf-tdTomato KI adult and neonatal mice were anesthetized with isoflurane using an inhalational anesthesia system[34–36]. Anesthesia was maintained using 1.0–1.5% isoflurane at a flow rate of ~0.2 L/min air. Anesthetized mice were placed on heating pads and monitored to maintain body temperature. The skin around the tibia was cut and the tibia was fixed to reduce the artifacts caused by the heartbeat and breathing to image the tibial bone marrow. A coverslip was placed on the exposed tibia. Intravenous injection of Qtracker 655 (Invitrogen) was conducted by retro-orbital injection (in adult mice) or superficial temporal vein injection using a pulled glass capillary (in neonatal mice) immediately prior to imaging. Intravital imaging of the tibial bone marrow was performed using a customized multiphoton laser-scanning upright microscope (Leica, SP8MP) with a 25× water immersion objective having a numerical aperture of 0.95 (Leica) and fiber oscillators that deliver pulses at 1070 nm (Coherent, Fidelity-2) and 920 nm (Spark lasers, Alcor-920). Fluorescence was collected by the 25× objective and directed to two HyD detectors (Leica) and one photomultiplier tube. Three-dimensional images of the bone marrow with a field of view of ~600 μm × 600 μm (512 × 512 pixels) were acquired every 5 μm.

**Image analysis to measure the velocity of tdTomato-positive cell movement**. General image processing was performed with ImageJ software[37]. Image registration was performed with the ImageJ plugin Correct 3D Drift[38] to correct the specimen drift in time-lapse images. The background noise was eliminated by masking the registered images with the binary tdTomato-positive cell images (mask images). Mask images were obtained from registered images with a 4–5 pixel-bandpass filtering, Otsu's thresholding, and manual postprocessing. The masked time-lapse images of tdTomato-positive cells were used to measure the velocity of the cell movement via cell tracking using the ImageJ plugin TrackMate[30].

**Histology and confocal microscopy**. Bones were fixed with 1% paraformaldehyde in phosphate buffer (PB) overnight at 4 °C and then incubated in 30% sucrose in PB for 1–2 days. Samples were embedded in O.C.T. Compound (Sakura Finetek), and then thick sections (more than 150 μm) of bone marrow were cut using a cryostat (Leica). Rabbit anti-DsRed antibody (for tdTomato) (632496, 1:500, Takara Bio Clontech), Alexa Fluor 488-conjugated anti-rabbit IgG (H + L) antibody (711-545-152, 1:1000, Jackson ImmunoResearch), PE-conjugated CD45.1 antibody (110708, 1:100, BioLegend), PE-conjugated CD45.2 antibody (109808, 1:100, BioLegend), Alexa Fluor 647-conjugated TER-119 antibody (116218, 1:100, BioLegend), and 4′,6-diamidino-2-phenylindole (DAPI) (10 μg/mL, Thermo Fisher Scientific) were used for histological staining. Bone sections were imaged using an SP8LS confocal microscope (Leica).

**RNA-seq**. For RNA-seq, 100 sorted HSCs were used to synthesize first-strand cDNA with PrimeScript RT reagent kit (TAKARA Bio Inc.) and not-so-random primers[4,39]. First-strand cDNA was treated with Klenow fragments (3′-5′ exo; New England Biolabs Inc.) and complement chains of not-so-random primers to synthesize the second-strand cDNA. The double-stranded cDNA was purified and an RNA-seq library was prepared and amplified using the Nextera XT DNA sample prep kit (Illumina Inc.). Prepared libraries were sequenced using Nextseq 500 (Illumina Inc.) and each obtained read was mapped to the reference sequence of mm10 using CLC genomic workbench, v11.0.0 (Qiagen). Expression levels were normalized and subjected to statistical analyses based on edgeR. Transcriptome data were subjected to GSEA using GSEA v4.0.3 software[40]. Gene sets were obtained from the Broad Institute database. Transcriptome data were repeatedly used for the analysis of the different gene sets.

**Statistical and reproducibility**. All results are presented as mean ± standard error. The significance of the differences was analyzed using a two-tailed Student's $t$-test (Fig. 4d) or a two-tailed Mann–Whitney $U$-test (Fig. 5b and Supplementary Fig. 2b) using Microsoft Excel and Statistical Package for the Social Sciences. Kolmogorov-Smirnov test was used to calculate the normality. Hedges' $g$ or $r$ was calculated to obtain the effect size. Hedges' $g$ was obtained by dividing the mean difference by the pooled standard deviation. The pooled standard deviation was calculated as the square root of $\{(n_1 - 1)SD_1{}^2 + (n_2 - 1)SD_2{}^2\}/\{(n_1 - 1) + (n_2 - 1)\}$, where $n_1$ and $n_2$ are the sample number of each group, and $SD_1$ and $SD_2$ are the standard deviations of each group. $r$ was obtained by dividing $z$ by the square root of $N$, where $z$ is the test statistic obtained by the $U$-test and $N$ is the sample size. All $p$ values <0.05 were considered statistically significant. The required sample size was calculated by R3.6.3[41] using power.t.test function with the following arguments: sig.level = 0.05, power = 0.8, $d$ = 1, and SD = 1 ($d$ = 1, SD = 1 were used to set the effect size as 1). Note that, for intravital imaging experiments, the sample number is lower than the required sample size due to technical difficulty.

**Reporting summary**. Further information on research design is available in the Nature Research Reporting Summary linked to this article.

## Data availability
Data were available upon reasonable request to the corresponding author (H.M.). RNA-seq data is available in the Gene Expression Omnibus repository (accession number: GSE16691).

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

## Acknowledgements
We would like to thank Hiromi Mizuno for their assistance regarding experiments. This research was supported by grants from the Ministry of Education, Culture, Sports, Science, and Technology (Grants-in-Aid for Scientific Research JP20K06876 and 20K09774).

## Author contributions
Y.T. and H.M. designed the study and wrote the first draft of the manuscript. Y.T., T.H., and H.M. designed the computational analysis of in vivo imaging data. Y.T., T.H., T.Y., T.U., K.A., M.H., M.S., H.T., T.I., T.S., and H.M. performed experiments/analyzed and interpreted data. T.Y., T.U., and T.S. revised the manuscript. All authors read and approved the final manuscript.

## Competing interests
The authors declare no competing interests.
