## [Peer Review File · Communications Biology]

This manuscript has been previously reviewed at another Nature Portfolio journal. This document only contains reviewer comments and rebuttal letters for versions considered at Communications Biology.

REVIEWERS' COMMENTS:

Reviewer #3 (Remarks to the Author):

In the manuscript by Takihara et al. the authors developed a novel imaging technique to detect hematopoietic stem cells in adult long bones and neonatal mice avoiding the damaging drilling of bones. The authors follow the colonization of neonatal HSCs in the bone marrow – and conclude that neonatal HSCs enter the marrow from the outside and settle on the blood vessel wall. The authors also conclude from their imaging data that neonatal HSCs are moving quicker when compared to adult HSCs.

The imaging technique that Takihara and colleagues have established is very exciting and has the potential to be used for several biological follow-up questions. The conclusions of the study are backup by the new data added by the authors. Still the authors should tone down the conclusions and discuss the challenge of their imaging method (e.g. challenge of monitoring more than 3 cells).

Comments for reviews:

Reviewer #3 (Remarks to the Author):

In the manuscript by Takihara et al. the authors developed a novel imaging technique to detect hematopoietic stem cells in adult long bones and neonatal mice avoiding the damaging drilling of bones. The authors follow the colonization of neonatal HSCs in the bone marrow – and conclude that neonatal HSCs enter the marrow from the outside and settle on the blood vessel wall. The authors also conclude from their imaging data that neonatal HSCs are moving quicker when compared to adult HSCs.

The imaging technique that Takihara and colleagues have established is very exciting and has the potential to be used for several biological follow-up questions. The conclusions of the study are backup by the new data added by the authors. Still the authors should tone down the conclusions and discuss the challenge of their imaging method (e.g. challenge of monitoring more than 3 cells).

We highly appreciate the reviewer’s helpful comments and thank for reviewing our study. As suggested, we have toned down the conclusions and discussed the challenge of the imaging method (lines 268-275).